# Parasitism and Suitability of *Aprostocetus brevipedicellus* on Chinese Oak Silkworm, *Antheraea pernyi*, a Dominant Factitious Host

**DOI:** 10.3390/insects12080694

**Published:** 2021-08-02

**Authors:** Jing Wang, Yong-Ming Chen, Xiang-Bing Yang, Rui-E Lv, Nicolas Desneux, Lian-Sheng Zang

**Affiliations:** 1Institute of Biological Control, Jilin Agricultural University, Changchun 130118, China; jingwang_90@126.com (J.W.); angusbio@126.com (Y.-M.C.); 2Subtropical Horticultural Research Station, United States Department of America, Agricultural Research Service, Miami, FL 33158, USA; 3Institute of Walnut, Longnan Economic Forest Research Institute, Wudu 746000, China; LRE390952@163.com; 4Institut Sophia Agrobiotech, Université Côte d’Azur, INRAE, CNRS, UMR ISA, 06000 Nice, France; nicolas.desneux@inra.fr; 5Key Laboratory of Green Pesticide and Agricultural Bioengineering, Guizhou University, Guiyang 550025, China

**Keywords:** egg parasitoids, parasitism, factitious host, fertilization, *Aprostocetus brevipedicellus*, *Antheraea pernyi*

## Abstract

**Simple Summary:**

The egg parasitoid *Aprostocetus brevipedicellus* Yang and Cao (Eulophidae: Tetrastichinae) is one of the most promising biocontrol agents for forest pest control. Mass rearing of *A. brevipedicellus* is critical for large-scale field release programs, but the optimal rearing hosts are currently not documented. In this study, the parasitism of *A. brevipedicellus* and suitability of their offspring on *Antheraea pernyi* eggs with five different treatments were tested under laboratory conditions to determine the performance and suitability of *A. brevipedicellus*. Among the host egg treatments, *A. brevipedicellus* exhibited optimal parasitism on manually-extracted, unfertilized, and washed (MUW) eggs of *A. pernyi,* and *A. brevipedicellus* offspring emerging from MUW eggs had high egg load. The results indicate that MUW eggs are optimal for the mass production of *A. brevipedicellus*.

**Abstract:**

*Aprostocetus brevipedicellus*, a eulophid gregarious egg parasitoid of lepidopterous pests, is a potential biological control agent for the control of many forest pests. A dominant factitious host, *Antheraea pernyi*, has been widely used for mass rearing several parasitoids in China. However, whether *A. pernyi* eggs are suitable for *A. brevipedicellus* rearing remains unclear. Here we evaluated *A. brevipedicellus* parasitism and fitness of their offspring on *A. pernyi* eggs with five different treatments, including manually-extracted, unfertilized and washed eggs (MUW), naturally-laid, unfertilized and washed eggs (NUW), naturally-laid, unfertilized, and unwashed (NUUW) eggs, naturally-laid, fertilized and washed eggs (NFW), and naturally-laid, fertilized and unwashed eggs (NFUW). The results showed that *A. brevipedicellus* could parasitize host eggs in all treatments but significantly preferred MUW eggs to other treatments. Moreover, *A. brevipedicellus* preferred unfertilized eggs to fertilized eggs and parasitized more washed eggs than unwashed. The pre-emergence time of parasitoid offspring emerging from fertilized eggs was shorter than that from unfertilized eggs. More parasitoid offspring emerged from unwashed eggs than that from washed eggs. The offspring emergence rate was high (>95%) and also female-biased (>85%) among all egg treatments. The egg load of female parasitoid offspring emerging from MUW and NUW eggs was 30–60% higher than the remaining treatments. Overall, MUW eggs of *A. pernyi* are the most suitable for the mass production of *A. brevipedicellus*.

## 1. Introduction

In China, forest insect pests have posed a serious threat to forest trees and agricultural crops. There are 5055 pest species accounting for damage in 11.4282 million ha of forestry in China [1]. The outbreaks of some pests often lead to a reduction in the yield of some economically important trees [2,3]. Management against crop and forest pests has mainly relied on insecticide applications [4,5]. However, long-term and extensive insecticide applications lead to environmental contamination and pest resistance development [6,7]. Furthermore, pesticide applications negatively impact non-target organisms such as beneficial arthropods (e.g., natural enemies of pests), leading to a resurgence of secondary pests, reducing biodiversity, and impacting overall ecosystem sustainability [7,8,9,10,11]. Therefore, it is necessary to develop effective and environmentally friendly control methods to control forest pests.

Biological control is a key method for pest management in forestry [12,13,14]. Among biological control agents that attack forest pests, parasitic wasps have received increasing attention due to their advantageous attributes such as host specificity, host feeding and parasitism, high efficiency, and easy mass rearing, etc. [15,16]. Eulophid parasitoids have been successfully used to control important forest pests in the United States [17], China [18], and worldwide [19,20]. Although biocontrol programs are promising, low abundance of natural enemies can lead to the failure of biological control programs, demonstrating a need to improve mass-rearing techniques for field augmentative release of natural enemies.

*Aprostocetus brevipedicellus* Yang and Cao (Eulophidae: Tetrastichinae) is an egg parasitoid first found in 2005 [21]. It can parasitize the eggs of lepidopterous pests such as *Caligula japonica* Moore, *Dendrolimu* spp., and *Lebeda nobilis* Walker and is one of the most promising biocontrol agents for forest pest control [22]. Many *Aprostocetus* parasitoids have successfully been used in forestry to control midge, beetle, and weevil pests [23,24,25,26,27,28].

Currently, mass rearing of *Aprostocetus* parasitoids relies on natural hosts for field release programs, such as mass rearing of *Aprostocetus hagenowii* Ratzeburg on the American cockroach *Periplaneta americana* L. for control [29]. To our knowledge, no mass rearing of *A. brevipedicellus* has been documented. The factitious eggs of Chinese oak silkworm *Antheraea pernyi* Guérin-Méneville are widely used for mass production of egg parasitoids such as *Trichogramma* and *Anastatus* because they are simple and cheap to mass-produce, and easy to transport [4,14,30,31,32,33]. Previous studies have demonstrated that the fertilization status of host eggs can be important for parasitism and fitness of the parasitoid offspring [34,35,36,37,38,39]. Generally, parasitoids prefer parasitizing fertilized host eggs [35,36,37,38]. However, this is not always the case [39]. The extracted eggs of *A. pernyi* from unfertilized females have been reported to use mass rearing of egg parasitoids [14]. A recent study demonstrated that *Trichogramma* parasitoids prefer parasitizing manually-extracted unfertilized washed *A. pernyi* eggs. *Trichogramma* offspring had increased fitness when reared on this factious host compared to natural hosts [34]. However, whether these unfertilized eggs can be used for *A. brevipedicellus* mass production remains unclear.

Additionally, host egg washing has been proved to influence parasitism preference and host orientation [34,40,41,42,43]. In this study, we explore the possibility of whether *A. pernyi* eggs can be used as a rearing host for *A. brevipedicellus*. We also evaluate the parasitism preference and fitness of *A. brevipedicellus* offspring on *A. pernyi* eggs under different host statuses. The objective of the current study was to determine the optimal *A. pernyi* egg treatment for mass production of *A. brevipedicellus*.

## 2. Materials and Methods

### 2.1. Parasitoids

*Aprostocetus brevipedicellus* was initially collected from parasitized eggs of Japanese giant silkworm *Caligula japonica* in walnut orchards in Kangxian (105–106° E, 32.9–33.7° N), Gansu Province, China, in 2017. The species were identified based on the morphological characteristics as described by Yao [21] and further confirmed by Dr. Gary Gibson in 2018 [5]. The voucher specimens of parasitoids were preserved in the Institute of Biological Control, Jilin Agricultural University, Changchun, Jilin province, China.

### 2.2. Host

Cocoons of Chinese oak silkworm, *A. pernyi*, were collected in Yongji City, Jilin Province, China, and then stored at 4 °C in the Institute of Biological Control, Jilin Agricultural University. After storage for 2–3 months, cocoons were transferred to an emergence room at 25 °C to allow adult emergence. Newly emerged adults (<6 h) were collected for the experiments.

### 2.3. Antheraea pernyi Egg Treatments

Egg treatments of *A. pernyi* were conducted to determine the optimal condition for the rearing of *Aprostocetus brevipedicellus*. The five treatments including (1) manually-extracted, unfertilized and washed eggs (MUW), (2) naturally-laid, unfertilized and washed eggs (NUW), (3) naturally-laid, unfertilized and unwashed eggs (NUUW), (4) naturally-laid, fertilized and washed eggs (NFW), and (5) naturally-laid, fertilized and unwashed eggs (NFUW). The preparation of the five egg treatments was similar to our previous study [34]. The MUW eggs were collected by dissecting the abdomen of unmated and mature female moths, and extracted eggs were subject to washing with distilled water immediately. Then washed eggs were air-dried under room temperature (about 1 h). When eggs dried, immature green eggs were removed, and the healthy eggs within 4 h after drying were used for the experiment. To prepare naturally-laid unfertilized eggs, the newly emerged female moths (<6 h) were collected and introduced to a screened cage (40 cm × 40 cm × 40 cm) to allow egg deposition. Our previous study found that 15% honey-water solution was suitable for a moth. Therefore, a 15% honey-water solution was also provided inside the cage for adult feeding. The cage was checked hourly, naturally-laid unfertilized eggs were divided into two groups; one group was washed with distilled water, air-dried under room temperature (NUW), and another group was not washed (NUUW). To prepare naturally-laid fertilized eggs, newly emerged female and male moths (<6 h) were collected and introduced to a screened cage (40 cm × 40 cm × 40 cm) to allow mating, with 15% honey-water solution provided as food. Newly laid fertilized eggs (<6 h) were divided into two groups, one was washed with distilled water and air-dried (NFW), and another group was not washed (NFUW). 0-day-old naturally-laid eggs were exposed to the parasitoids.

### 2.4. Suitability of A. pernyi Egg with Different Treatments on Parasitism by A. brevipedicellus

#### 2.4.1. No-Choice Test

A no-choice experiment of *A. brevipedicellus* on treated *A. pernyi* eggs was conducted under laboratory conditions (25 ± 1 °C, 70 ± 5% RH, and 14 L: 10 D). Newly emerged *A. brevipedicellus* females (<1 h old) were collected and introduced with males in glass tubes provided with 10% honey-water solution as food (2.5 cm × 12 cm, diameter × height) to allow mating for 3 days before use in experiments. Based on our observations, one *A. brevipedicellus* female can parasitize up to 20 *A. pernyi* eggs within 24 h. Therefore, mated female parasitoids were individually exposed to 40 host eggs in a no-choice test for 24 h to allow a surplus supply of hosts. For each treatment, forty host eggs were glued onto a paper card (0.5 cm × 5.0 cm) using nontoxic glue and placed inside a glass tube (1.0 × 7.5 cm, diameter × height). One three-day-old, mated female *A. brevipedicellus* was also introduced into the glass tube containing an egg card. Our previous study found that a 10% honey-water solution was suitable for *A. brevipedicellus*. Therefore, a 10% honey-water solution was provided as food during the test. After 24 h, *A. brevipedicellus* was removed from the glass tube. The host eggs on the card were then cut out from the card and held individually in a glass tube, and maintained in an incubator under the conditions as described above. All parasitized eggs were checked daily until the emergence of all adults. After there was no further adult emergence, host eggs were individually dissected under a stereomicroscope to check the number of dead wasps left inside the chorion. For each egg treatment, the number of parasitized eggs (number of the host eggs with emergence hole + number of host eggs without emergence hole but containing parasitoids), pre-emergence time (d) (the number of days from exposure of host eggs to the parasitoid to the adult offspring emerging from host egg), the number of emerged adults per host egg (dead adults inside host eggs were excluded), the emergence rate (the number of eggs with emergence holes/total number of parasitized eggs × 100), percentage of female progeny (number of emerged females/total number of emerged females and males × 100), the number of dead parasitoids left inside the eggs, female offspring body size (left hind tibia length of female adults, HTL), and the egg load per female offspring were recorded. For HTL, thirty female adults from each treatment were selected randomly, and HTL was measured using an ultramicroscope (Keyence VHX-2000). To record the egg load per female offspring, thirty newly-emerged females were randomly selected from each treatment and then dissected under a stereomicroscope to examine the ovary to document the number of eggs. For each egg treatment, 30 female *A. brevipedicellus* were tested.

#### 2.4.2. Choice Test

A choice experiment was tested under laboratory conditions (25 ± 1 °C, 70 ± 5% RH and 14 L: 10 D) to determine the host preference of *A. brevipedicellus* for *A. pernyi* eggs among the five treatments previously listed. One three-day-old and mated female *A. brevipedicellus*, as described above, was introduced into a glass tube (1.0 × 7.5 cm, diameter × height) containing an egg card. The egg card carried forty *A. pernyi* eggs from all egg treatments, with eight eggs per treatment, and was randomly placed on the paper card (0.5 cm × 5.0 cm). 10% honey-water solution was also provided as food. After 24 h, parasitoids were removed, and host egg cards were cut out individually and kept in an incubator as described above for posttreatment observation. After 6 days, the eggs were examined under a stereomicroscope to document the number of parasitized eggs of each treatment. This experiment was replicated 30 times.

### 2.5. Statistical Analysis

For the no-choice bioassay, a one-way analysis of variance (ANOVA) was conducted to determine the effect of treatment on the number of parasitized eggs, pre-emergence time, number of emerged adults per egg, emergence rate, number of dead wasps left inside per egg, percentage of female progeny, hind tibia length and egg load of per female offspring. Tukey’s honestly significant difference (HSD) test was used to compare means at *p* < 0.05. All data were subject to a normality test (Shapiro–Wilk test) prior to ANOVA. All percentage data were arcsine square-root-transformed prior to the Shapiro–Wilk test. The analysis was performed on the transformed data, and untransformed means ± SE were presented. For the choice test, data were analyzed using the Friedman non-parametric analysis to determine the effect of treatment on the number of parasitized eggs. All data analyses were performed using SPSS 23.0 (SPSS Inc., Chicago, IL, USA).

## 3. Results

### 3.1. Suitability of A. pernyi Egg with Different Treatments on Parasitism by A. brevipedicellus

#### 3.1.1. No-Choice Test

Fertilization significantly affected the number of eggs parasitized (*F*_1,116_ = 8.10, *p* = 0.0052) regardless of washing or not. However, the interaction between fertilization and washing treatments did not affect the number of parasitized eggs (*F*_1,116_ = 0.81, *p* = 0.3686). *Aprostocetus brevipedicellus* successfully parasitized *A. pernyi* eggs in all egg treatments, and there was a significant difference in the number of parasitized eggs among treatments (*F*_4,145_ = 130.07, *p* < 0.0001) (Figure 1A). The manually-extracted egg treatment resulted in the maximum number of parasitized eggs, which was 63–242% higher than the remaining treatments.

#### 3.1.2. Choice Test

In the choice test, MUW eggs of *A. pernyi* resulted in the maximum number of parasitized eggs, which was 259–2533% higher than the remaining treatments (*χ*^2^ = 101.41, df = 4, *p* < 0.0001) (Figure 1B). *Aprostocetus brevipedicellus* parasitized 578% more NUW eggs than NUUW eggs (*Z* = 4.25, *p* < 0.0001), and parasitized 633% more NFW eggs than NFUW eggs (*Z* = 3.51, *p* < 0.0001). The number of parasitized eggs was equivalent between NUUW and NFUW treatments (*Z* = 0.37, *p* = 0.7130).

### 3.2. Suitability of A. pernyi Egg with Different Treatments on Development of A. brevipedicellus Offspring

#### 3.2.1. Pre-Emergence Time

The pre-emergence time of *A. brevipedicellus* offspring was 6–10% shorter on NFW and NFUW treatments compared with MUW, NUW, and NUUW treatments (*F*_4,735_ = 36.83, *p* < 0.0001) (Figure 2A). 

#### 3.2.2. Emergence of *Aprostocetus brevipedicellus*

The number of emerged *A. brevipedicellus* adults per *A. pernyi* egg was 10–31% higher in the NFUW treatment compared with the remaining egg treatments (*F*_4,735_ = 26.38, *p* < 0.0001) (Figure 2B). MUW and NUW egg treatments resulted in about seven adults per host, significantly lower than the remaining treatments. Treatment had a significant effect on the percentage of female progeny (*F*_4,735_ = 3.14, *p* = 0.0141). *Aprostocetus brevipedicellus* offspring emerging from MUW egg treatment was 3% more female-biased than the NUW egg treatment (Figure 2C). Emergence rate did not significantly vary between all egg treatments (*F*_4,761_ = 0.28, *p* = 0.8940), and ranged from 95% to 97%. Similarly, there were no significant differences in the number of dead parasitoids per host egg between all egg treatments (*F*_4,761_ = 0.53, *p* = 0.7170). The number of dead parasitoids per egg was the highest in those from NFUW host eggs (0.35), followed by NUUW (0.23), NFW (0.22), MUW (0.17), and NUW host eggs (0.17). 

#### 3.2.3. Female Hind Tibia Length and Egg Load of *Aprostocetus brevipedicellus* Offspring

Hind tibia length of *A. brevipedicellus* female offspring ranged from 410.50 μm to 428.10 μm, and no significant differences were found in the hind tibia length among all egg treatments (*F*_4,145_ = 0.80, *p* = 0.5277). Female offspring emerging from MUW egg treatment resulted in the longest hind tibia length, followed by NUW, NFW, NUUW, and NFUW. *Aprostocetus brevipedicellus* female offspring emerging from MUW egg treatment resulted in maximum egg load, which was 2–12% higher than the remaining treatments (*F*_4,145_ = 8.04, *p* < 0.0001) (Figure 3). 

## 4. Discussion

The large eggs of the Chinese oak silkworm, *A. pernyi*, have been demonstrated as an excellent substitute host for mass-production for several egg parasitoids, especially *Trichogramma* and *Anastatus* [14,30,33,34]. In this study, we evaluated the parasitism of five *A. pernyi* egg treatments by *A. brevipedicellus,* and the fitness of resulting *A. brevipedicellus* offspring. The results showed that *A. brevipedicellus* successfully parasitized *A. pernyi* eggs in all egg treatments with varied parasitism and host suitability performance. Our study indicated that the MUW *A. pernyi* eggs showed great potential for mass production of *A. brevipedicellus*, since the results showed that this treatment resulted in the maximum number of parasitized host eggs in both the choice and no-choice experiments and high egg load in offspring from this treatment.

Previous studies have demonstrated that the host quality [14,44,45] and fertilization of host eggs can affect the parasitism rate, with most species preferring to parasitize fertilized eggs [35,36,37,38]. However, our results showed that *A. brevipedicellus* could recognize both fertilized and unfertilized host eggs and preferred parasitizing unfertilized eggs. Similar findings were also reported for *T. japonicus* and *T. dendrolimi*, which prefer unfertilized *H. halys*, and *A. peryni* eggs for mass rearing [34,39]. In addition, the present study found that the washing of host eggs influenced parasitoid preference, and *A. brevipedicellus* parasitized significantly more washed eggs than unwashed. Wang et al. found the same results for *T. dendrolimi* parasitizing *A. pernyi* eggs [34]. This may be attributable to changes in volatile profiles of *A. pernyi* eggs after washing treatments.

Furthermore, the washing treatment may soften and thin the egg chorion due to the friction between the eggs during washing. However, further studies are required to verify this phenomenon. Among the washed *A. pernyi* eggs, *A. brevipedicellus* parasitized significantly more manually-extracted eggs than naturally-laid eggs. The difference in preference may be due to the immature chorion of eggs, which is easier for *A. brevipedicellus* oviposition. The early manual extraction of eggs may have terminated the tanning process related to hardening the egg chorion [44]. Previous studies have demonstrated that the hardness and thickness of *A. pernyi* chorion is a limiting factor for some parasitoids [33,34]. Results presented here showed that a hard egg chorion can be an obstacle to parasitism and is not conducive to mass production.

The performance of *A. brevipedicellus* offspring was also affected by the fertilization of host eggs [34,35,37,38]. This study found that the pre-emergence time of *A. brevipedicellus* developing in fertilized eggs of *A. pernyi* was significantly shorter than that in unfertilized eggs. Similarly, Yang et al. reported that three *Trichogramma* species had shorter developmental time in fertilized eggs of *Chilo suppressalis* Walker than those in nonfertilized [37]. The embryonic development of fertilized host eggs can significantly influence the fitness of parasitoids [46]. The difference in the pre-emergence time between fertilized and unfertilized eggs seen in the current study may be attributable to the process of embryonic development, which increased accessibility of various nutrients within the host egg before exposure to the parasitoid. This increase in nutrient availability may have led to faster development of *A. brevipedicellus* offspring on fertilized eggs than unfertilized eggs. Other factors may also be involved, e.g., egg deposition period [47], but they were not measured in the present study.

Many female parasitoids can assess host quality and quantity to determine suitability for offspring development [46,47,48,49]. Our data showed that the number of emerged adults per host egg was significantly affected by the fertilization and washing treatment, and more *A. brevipedicellus* adults emerged per host egg from fertilized eggs than from unfertilized eggs. Similarly, egg parasitoids *Gonatocerus morrilli* Howard and *Telenomus coloradensis* Crawford parasitizing on *Homalodisca vitripennis* Germar eggs had higher mortality on unfertilized host eggs [35]. This is likely due to the increased accessibility of various nutrients associated with the fertilized *A. pernyi* eggs. We suspect that embryonic development of fertilized *A. pernyi* egg improves accessibility to nutrients in the host egg for the parasitoid. Therefore, when parasitizing fertilized eggs, the parasitoids might lay the larger number of eggs per host egg. A recent study on *Trichogramma* parasitizing *A. pernyi* eggs also indicated that the number of emerged adults per fertilized host egg was significantly higher than per unfertilized host egg [34]. There was a significantly higher number of emerged adults per egg on unwashed eggs than that on washed for the host egg washing treatments. We suspect that, compared with the washed eggs, it is difficult for parasitoids to parasitize unwashed eggs with the thicker and harder chorion. Therefore, parasitoids will expect to lay a larger number of offspring inside if they successfully parasitize an unwashed egg. This may be a parasitic strategy when parasitoids are confronted with disadvantage conditions [50,51]. 

Results showed that *A. brevipedicellus* offspring was strongly female-biased (>85% females) in all treatments. The percentage of female progeny under the natural field conditions was 77% [5], and smaller than that under laboratory conditions. Previous studies have demonstrated that it is preferable to have a female biased sex ratio of the parasitoids in biocontrol programs [29,52]. There is often a positive relationship between fitness and body size [46,53,54,55], and the body size of parasitoids is often positively correlated to their fecundity [56,57]. For example, Wang et al. found that the body size of female *Ooencyrtus kuvanae* Howard positively affects its fecundity [58]. Host quality can also influence the egg load of parasitoids [14,59]. Li et al. reported that the number of emerged *Oomyzus sokolowskii* Kurdjumov adults per host was negatively correlated with egg load [60]. Although there were no significant differences in parasitoid body size between treatments, female offspring emerging from MUW and NUW eggs had the highest egg load. NUW resulted in 39–61% less parasitism than the MUW treatment, suggesting that MUW *A. pernyi* eggs are the most optimal host for the mass rearing of *A. brevipedicellus*.

## 5. Conclusions

In conclusion, MUW eggs of *A. pernyi* were most suitable for oviposition based on host preference, parasitism, and parameters of offspring fitness. Our results provide evidence to support that MUW eggs of *A. pernyi* are desirable for mass production of *A. brevipedicellus*. They also provide the possibility for large-scale application of *A. brevipedicellus* to control the forest pests effectively. However, further studies are needed to optimize the rearing ratio of *A. brevipedicellus* to MUW eggs and their biocontrol efficiency against target pests in field release programs.

## Figures and Tables

**Figure 1 insects-12-00694-f001:**
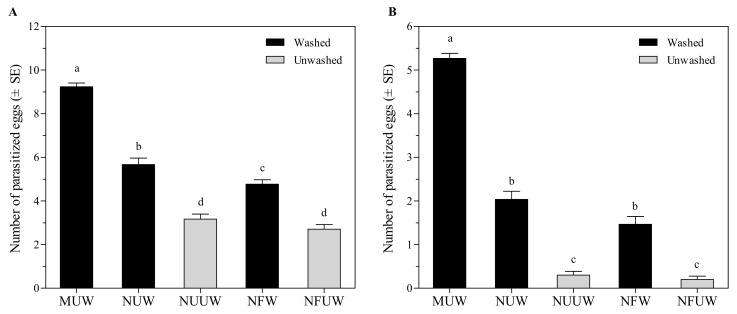
Number of parasitized *Antheraea pernyi* eggs by *Aprostocetus brevipedicellus* with different treatments in no-choice test (**A**) and the choice test (**B**). MUW: manually-extracted, unfertilized and washed eggs; NUW: naturally-laid, unfertilized and washed eggs; NUUW: naturally-laid, unfertilized and unwashed eggs; NFW: naturally-laid, fertilized and washed eggs; NFUW: naturally-laid, fertilized and unwashed eggs. In the no-choice test, different lower-case letters on top of bars indicate significant differences among egg treatments based on Tukey’s multiple comparison (HSD) test (*p* < 0.05). In the choice test, different lower-case letters on bars indicate significant differences among egg treatments based on Friedman’s significant difference test (*p* < 0.05).

**Figure 2 insects-12-00694-f002:**
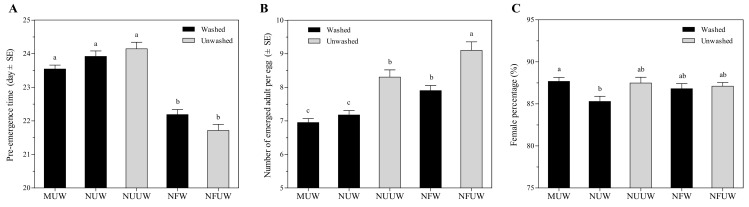
Pre-emergence time (**A**), the number of emerged adults per egg (**B**), and female progeny percentage (**C**) of *Aprostocetus brevipedicellus* on *Antheraea pernyi* eggs with the five treatments. MUW: manually-extracted, unfertilized and washed eggs; NUW: naturally-laid, unfertilized and washed eggs; NUUW: naturally-laid, unfertilized and unwashed eggs; NFW: naturally-laid, fertilized and washed eggs; NFUW: naturally-laid, fertilized and unwashed eggs. Different lower-case letters on bars indicate significant differences among egg treatments based on Tukey’s multiple comparison (HSD) test (*p* < 0.05).

**Figure 3 insects-12-00694-f003:**
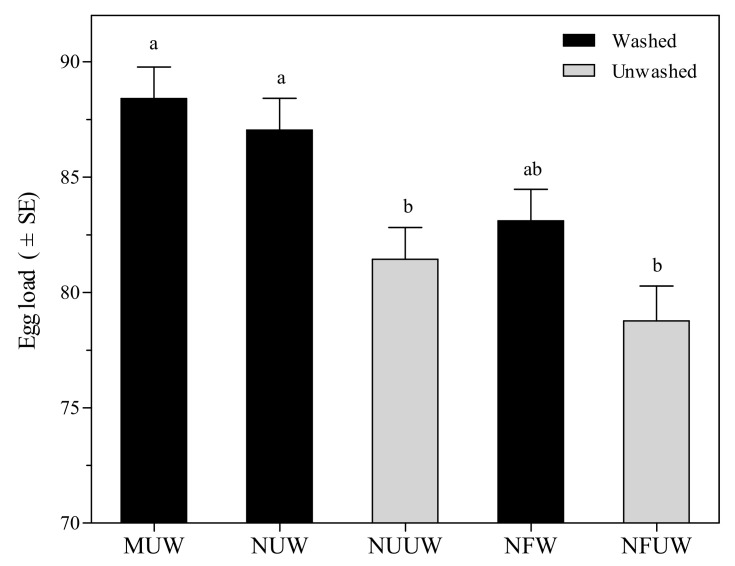
Egg load of *Aprostocetus brevipedicellus* offspring from *Antheraea pernyi* eggs with five treatments. MUW: manually-extracted, unfertilized and washed eggs; NUW: naturally-laid, unfertilized and washed eggs; NUUW: naturally-laid, unfertilized and unwashed eggs; NFW: naturally-laid, fertilized and washed eggs; NFUW: naturally-laid, fertilized and unwashed eggs. Different lower-case letters on bars indicate significant differences among egg treatments based on Tukey’s multiple comparison test (HSD) (*p* < 0.05).

## Data Availability

Not applicable.

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
