# Peer review of "Parasitism and Suitability of Aprostocetus brevipedicellus on Chinese Oak Silkworm, Antheraea pernyi, a Dominant Factitious Host"

_insects, 2021, doi:10.3390/insects12080694_

Round 1
Reviewer 1 Report
Please see attachment.

Author Response
Reviewer 1:
This is a worthwhile study investigating the most optimal A. pernyi egg for mass rearing of A. brevipedicellus. However, the manuscript needs significant revision, particularly in the areas of (1) reducing the number of detailed references to others work; (2) improving the English used; and (3) condensing the number of figures that represent the results.
Response: Thanks for the active comments on our work. According to your three suggestions, we do the following revisions and responses.
(1) reducing the number of detailed references to others work: According to the suggestions, we delete 8 references as followed.
Consoli, F.L.; Vinson, B.S. Host regulation and the embryonic development of the endoparasitoid Toxoneuron nigriceps (Hymenoptera: Braconidae). Comp. Biochem. Physiol. B: Biochem. Mol. Biol. 2004, 137, 463–473.
Fischbein, D.; Corley, J.C. Classical biological control of an invasive forest pest: a world perspective of the management of Sirex noctilio using the parasitoid Ibalia leucospoides (Hymenoptera: Ibaliidae). Bull. Entomol. Res. 2015, 105, 1–12.
Halfpapp, K. Introduction of Tetrastichus brontispae for Control of Brontispa longissima in Australia. Proceedings of the Sixth Workshop for Tropical Agricultural Entomology, Darwin, Australia, 11–15 May 1998; Technical Bulletin Department of Primary Industry and Fisheries: Northern Territory of Australia, 2001; 288, 5960.
Han, P.; Niu, C.Y.; Biondi, A.; Desneux, N. Does transgenic Cry1Ac+CpTI cotton pollen affect hypopharyngeal gland development and midgut proteolytic enzyme activity in the honey bee Apis mellifera L. (Hymenoptera, Apidae)? Ecotoxicology 2012, 21, 2214–2221.
Harvey, J.A.; Sano, T.; Tanaka, T. Differential host growth regulation by the solitary endoparasitoid, Meteorus pulchricornis in two hosts of greatly differing mass. J. Insect Physiol. 2010, 56, 1178–1183.
Huang, D.Z.; Liu, H.F.; Wang, Z.G.; Yang, Z.Q.; Li, H.P. Study on reproductive biology of Aprostocetus prolixus on eggs of Apriona germarii (Coleoplera: Cerambycidae). Sci. Silvae Sinicae 2005, 41, 195–200. (In Chinese)
Qiu, L.F.; Sun, X.G.; Sun, S.J.; Gao, J.Y. Host-searching behavior of the Aprostocetus fkutai Miwa et Sonan. J. Environ. Entomol. 2003, 25, 24–27. (In Chinese)
van Lenteren, J.C.; Bolckmans, K.; Köhl, J.; Ravensberg, W.J.; Urbaneja, A. Biological control using invertebrates and microorganisms: plenty of new opportunities. Biocontrol 2018, 63, 39–59.
Zhang, S.; Kong, X.B.; Wang, H.B.; Zhou, G.; Yu, J.X.; Liu, F.; Zhang, Z. Sensory and immune genes identification and analysis in a widely used parasitoid wasp Trichogramma dendrolimi (Hymenoptera: Trichogrammatidae). Insect Sci. 2016, 23, 417–429.
- improving the English used.
The reviewer 2 and you have given lots of valuable revisions on English, we accept all of your suggestions. Meanwhile, we also improved English as far as we can. Please refer to text.
- condensing the number of figures that represent the results.
Done as suggested. We checked the figures, it is suitable to condense Figure 1 and 2 into 1 figure.
Abstract:
Line 20: There is no need to put “The” before scientific names or, for example, before “lepidopterous”. Please amend throughout the manuscript.
Response: Done. Revised as suggested throughout the manuscript.
Line 20: Suggest to write “…..eulophid gregarious egg parasitoid….
Response: Done.
Line 22: change “egg is a suitable” to “eggs are a suitable”
Please list the treatments in the abstract after “five different treatments”.
Response: Done, please see lines 29-32.
Line 28: change to “unwashed eggs that from washed eggs”.
Response: Done.
Line 29: Change to “Female parasitoids emerging from MUW….”
Response: Done.
Line 30: Suggest to write “…the most suitable….
Response: Done.
Introduction:
Line 33: change to “a serious threat”
Response: Done.
Line 35: what are the economic losses – do you mean in reduction in crop quality and yield? What is the ecological damage?
Response: We revised the sentence to ‘The outbreaks of some pests often lead to reduction in yield of some economically important trees and biodiversity’.
Line 35: Remove “For a long period”. Change to “Management against forest pests have mainly relied on….”
Response: Done.
Line 36: Change to “However, long-term and extensive insecticide applications have resulted in environmental contamination and pest…..”
Response: Done.
Line 39: Change to “Therefore, there is a need to develop…”
Response: Done.
Line 41: There is no need to put “the” in front of “biological control”. Change to “Biological control is a key method for pest management in forestry. Among the biological control agents that attack forest pests, parasitic wasps have received increasing…..”
Response: Done.
Line 43: ‘Host searching’ probably wouldn’t be considered an advantageous attribute. Maybe you mean ‘host specificity’ which is advantageous over generalize predators because parasitoids only attack a specific pest species and therefore have less non-target effects. Host feeding and parasitism are advantageous because they kill the pest. Consumption of host nutrition is not an advantage, this is a biological attribute. What do you mean by ‘high efficiency’? Do you mean that gregarious parasitoids can produce many offspring per host?
Response: It is revised as ‘parasitic wasps have received increasing attention due to their advantageous attributes such as host specificity, host feeding and parasitism, high efficiency, and easy mass rearing etc.’
Lines 44-55: There are too many references and examples here. They also jump from China examples to the rest of the world and the US. Suggest to only include a few examples that use parasitoid species that are in the same family as the parasitoid you are studying. If there are no other Eulophid parasitoid examples in the literature, then just select egg parasitoids attacking forest pests in the same family as your study insect. It’s great you’ve done such a thorough literature review, but it’s unnecessary to include them all in a scientific paper. Its ok to just state for example, “Eulophid egg parasitoids have been successfully used to control important forest pests in United States (references), China (references) and worldwide (references).
Response: Thanks for the comment, we accept your revision suggestion. Done.
Lines 56-59: I am unsure what these sentences are trying to say. Are you trying to say that low abundance of natural enemies leads to the failure of biological control programs, and that there is a need for improving mass-rearing techniques for augmentative release of natural enemies?
Response: Yes, it is what we want to say. The sentence is revised as ‘Although biocontrol programs are promising, low abundance of natural enemies were not sufficient to control the pest in biological control programs, and that there is a need for improving mass-rearing techniques for field augmentative release of natural enemies, and that there is a need for improving mass-rearing techniques for field augmentative release of natural enemies.’
Line 60: No need to state the genus twice in the same sentence. Where was it first found – is this in China?
Response: Thanks. We deleted “in Aprostocetus genus and was”. The sentence is revised as ‘Aprostocetus brevipedicellus Yang and Cao (Eulophidae: Tetrastichinae) is an egg parasitoid first found in 2005’
Line 63: Remove “Aprostocetus genus is a large group of family [41].” The genus is the genus, not the family – this part of the sentence is confusing and not needed. Start sentence “Many Aprotocetus…” Also, no need to give lots of detail about all of the examples. Can simply state “Many Aprotocetus parasitoids have successfully been used in forestry to control midge, beetle, and weevil pests (references).
Response: Thanks for the comment, we accept your revision suggestion. Done.
Line 69: Change to “…parasitoids mainly relies on natural…..”
Response: Done.
Line 71: This sentence and the last sentence both start with “currently”. Change to “Currently, no mass rearing of A. brevipedicellus has been documented.”
Response: Done.
Line 75: Change to “A recent study…”
Response: Done.
Line 79: Change to “Generally, parasitoids prefer parasitizing fertilized host eggs (list references), however, this is not always the case (reference).” There is no need to include details of specific examples here.
Response: Thanks for the comment, we accept your revision suggestion. Done.
Line 84: remove “the” before “host egg”
Response: Done.
Methods:
Why was 10% and 15% honey water solution used for different experiments? Why wasn’t this standardized?
Response: Thanks for the comment. Our previous study found that 10% honey-water solution is suitable for the parasitoids and 15% honey-water solution is suitable for moth. So, 10% honey-water solution was used for the parasitoids, 15% honey-water solution was used for moth.
Line 98: Chanage to “…were transferred to an emergence room at 25℃ to allow adult emergence. Newly emerged (<6 h?) adults were …..”
Response: Done.
Line 101: Why did you not include the treatments “manually-extracted unfertilized unwashed eggs”? Could change to “The five egg treatments included … (list treatments). No need to have “…were tested for A. brevipedicuellus” on the end of this sentence.
Response: Thanks for the question. When manually extracting the unfertilized eggs, the eggs were covered by a layer of muscus and all eggs were stick together, and they would turn black after being air dried. So the manually-extracted unfertilized unwashed egg was not considered as a treatment in this study. The sentence is revised as suggested.
Line 111: “…15% honey-water solution…”
Response: Done.
Line 114: No need to state in words AND also have symbols for male and female. Just choose one.
Response: Done.
Line 117: What do you mean by “suitability of Aprostocetus”? This should say “… on parasitism by Aprostocetus”. Or “The suitability of A. pernyi egg treatments for parasitism by A. brevipedicellus”.
Response: It is revised as ‘Suitability of A. pernyi egg with different treatments on parasitism by A. brevipedicellus’.
Line 120: Did you use fluorescent lighting?
Response: Yes, we used fluorescent lighting. In this experiment, the climate incubator is equipped with fluorescent light tubes, we set time of light and dark to 14:10.
Line 121: “glass tubes”; “10% honey-water solution”.
Response: Done.
Line 122: “… to allow mating for 3 days prior to use in experiments”.
Response: Done.
Line 123: “… to allow surplus supply of hosts.”
Response: Done.
Line 124: Remove “to start the no-choice test”. Replace instead with “For each treatment, forty host eggs….”
Response: Done.
Line 126: Change to “One three-day old, mated female A. brevipedicellus… glass tube containing an egg card”. Also, there is no need to have “the” in front of “10% honey solution.” Could write “A drop of 10% honey water solution was….”
Response: Done.
Line 130: “the chorion”.
Response: Done.
Line 133: How was pre-emergence time calculated? The number of days from exposure of eggs to the parasitoid to the emergence of the adult offspring?
Response: We add the calculation method in the text: the number of days from exposure of host eggs to the parasitoid to the adult offspring emerging from host egg.
Line 133-140: The 100% does not go at the beginning of the equation. Suggest to write as “(The number of eggs with emergence holes/total number of parasized eggs) x 100”
Response: Done.
Line 136: “the number of eggs loaded per female offspring” does not make sense. Do you mean “egg load of female offspring”?
Response: Yes, “the number of eggs loaded per female offspring ” is changed to “egg load of female offspring”.
Line 138: “…. selected randomly and HTL measured using an ….”
Response: Done.
Line 143: State “… among the five treatments previous listed”.
Response: Done.
Line 146: Change to “…were randomly placed on the paper…”
Response: Done.
Line 151-159: You need to state the analyses was conducted to determine the parameters “between treatments”. You have stated that some data were arcsine square-root transformed. How were the means calculated in your results? Are these calculated from raw data? Or were they estimated by the model and back-transformed?
Response: Based on the comments, the related sentence is revised as ‘one-way analysis of variance (ANOVA) was conducted to analyze the number of parasitized eggs, pre-emergence time, number of emerged adults per egg, emergence rate, number of dead wasps left inside per egg, percentage of female progeny, hind tibia length and egg load per female offspring between treatments’. We add a sentence ‘The analysis was performed on the transformed data and untransformed means±SE were presented.’
Results:
The methods and results section should have similar headings for the sections.
Response: Done.
You could include all the statistical results (F values, df and P values) for all experiments in one table to remove these from the text.
Response: Thanks for the comment. Considering no-choice and choice tests, we keep statistical results (F values, df and P values) in the text.
The results section doesn’t give any specific details about the trends seen. For example, instead of saying “significantly higher” or “more”, state how much higher - “20% higher”.
Response: Thanks for your suggestion. Here we tend to generally describe the results, so we use “significantly higher” or “more” between different treatments. Detailed results can be found from presented figures.
There needs to be consistency between the results sections. The no choice tests don’t have the treatments listed from greatest to least, but the Choice tests have. These results can be easily seen from the figures so is unnecessary. Perhaps you can state which treatment resulted in maximum number of parasitoid eggs, and state that this was 28-60% higher than the remaining treatments.
Response: According to the comments, we add ‘Generally, Manually-extracted egg treatment resulted in maximum number of parasitized eggs, and that was 62.94-241.97% higher than the remaining treatments.’ for no-choice test, and ‘Generally, manually-extracted egg treatment resulted in maximum number of parasitized eggs, and that was 259.09-2533.35% higher than the remaining treatments (χ2 = 101.41, df = 4, P < 0.0001) (Figure 1B).’ for choice test.
Line 193: You need to include “between treatments” on that first sentence.
Response: Done. “In the naturally-laid egg treatments,”is added as the first sentence.
Line 195: You need to include “between treatments” after “female progeny”.
Response:Done. We add “between MUW eggs and NUW eggs” after “female progeny”.
The figures can be condensed into two figures, or one table. The legend on the figures should be the five different egg treatments, and the x-axis would have the variable (eg. parasitism in no choice experiment, parasitism in choice experiment, number of emerged adults per egg, etc.)
Response: Done. Based on the comments, we condense Figure 1 and 2 into one figure, and we did not change the left figures.
Results for Emergence rate, Dead parasitoids and HTL are not significant and can be removed from the figures. Simply state in the results section the range of these variables and the test statistics showing they are insignificant. Eg., HTL ranged from 430-415 um ± SE and did not significantly vary between egg treatments (test statistics).
Response: Thanks for the suggestions. Although Emergence rate, Dead parasitoids and HTL are not significant between different treatments, it is clear to show the results with presented figures. So we keep these figures in the paper.
Discussion
Line 209: a comma is needed after “silkworm” and after “pernyi”
Response: Done.
Line 211: Do you mean “parasitism of five A. pernyi egg treatments by A. brevipedicellus and the fitness of resulting A. brevipedicellus offspring?
Response: Yes, thanks. We revised the sentence per your suggestion to “…we evaluated the parasitism of five A. pernyi egg treatments by A. brevipedicellus and the fitness of resulting A. brevipedicellus offspring and the…”.
Line 215: Suggest to change to “……A. brevipedicellus, since results showed that this treatment resulted in maximum number of parasitized host eggs in both the choice and no choice experiments and high egg load in offspring from this treatment”
Response: Done.
Line 216: Figure doesn’t show that MUW treatment resulted in the “highest” egg load. MUW and NUW are statistically equivalent. Perhaps change highest to high.
Response: Done.
Line 220: Remove the sentence starting with “These results…”
Response: Done.
Line 221: Change to “In addition, the present study found that parasitoid preference was influenced….”
Response: Done.
Line 230: Add “have” after “previous studies”.
Response: Done.
Line 231: “A hard egg chorion can be an obstacle for parasitism and is not conducive to mass production”
Response: Done.
Line 233: Are the references 55, 56, 58 and 59 specifically related to A. brevipedicellus? Or are these your results?
Response: Yes, the references 55, 56, 58 and 59 are specifically related to A. brevipedicellus.
Line 236: Walker has a capital W
Response: Done.
Line 236-242: This section is difficult to understand.
Response: The other reviewer suggested to revise as “the process of embryonic development that preceded parasitism increased accessibility of various nutrients within the host egg for the parasitoid”
Line 243: “Many female parasitoids can assess host quality and quantity to determine suitability for offspring development”.
Response: Done.
Line 245: Remove “were”
Response: Done.
Line 250: Suggest to remove sentence starting with “Thus,…..”
Response: Done.
Line 253: “… parasitoids possibly spend longer ovipositing.”
Response: Done.
Line 256-266: This paragraph changes too much from one idea to another. Suggest to remove first sentence. Start with “Results showed that A. brevipedicellus offspring was strongly female biased (<85%) in all treatments, whereas, a previous study has shown that the percentage of female progeny under natural field conditions was 77% [5]. It is preferable to have female biased sex ratio in biocontrol programs using parasitoids.
Response: Done.
Line 265: Change to “Female offspring emerging from MUW eggs had the highest egg load which suggests that MUW eggs are…..”
Response: Done.
References:
Suggest to only include references that relate to the Family of the study parasitoid and/or the Family of the study pest to help reduce the number of references.
Response: According to your suggestion, we delete 8 references, see above responses.
Reviewer 2 Report
The manuscript is interesting and generally clearly written. The results may become very useful in developing a mass-rearing program. I made a variety of edits on the manuscript to improve clarity, and I hope the authors find those useful. I did not review the literature cited in depth to make sure all were cited, or formatted correctly.
I also have some questions/comments about the manuscript (these are also in the comments on the pdf):
Line 28: It would useful to outline the treatments in the abstract, rather than making the reader try to extract them from the results text.
Lines 67-69: I don’t understand this sentence. Can you clarify your points?
Lines 122-123: How long after manual removal were eggs exposed to the parasitoids? Similarly, how long after washing were eggs exposed, and what was the post-oviposition age of the naturally deposited eggs at the time they were exposed to the parasitoids? What are the optimal host egg ages for parasitism by this parasitoid?
Lines 122-131: How did egg size compare within the manually extracted eggs? Were they all similar in size? And how did the manually extracted egg size compare with eggs naturally oviposited?
Lines 167-169: Were there any efforts made to assess number of parasitoids per parasitized egg to see if the parasitoids modified their oviposition patterns in response to the choice options?
Line 237: There were only significant differences in % female offspring between MUW and NUW.
Lines 280-282: How was this softening recognized? Did you measure behavioral elements of oviposition to identify this?
Lines 295-297: This idea depends on the status of the host egg after oviposition. If the parasitoid essentially stops development after oviposition, as is common among egg parasitoids, then competition is not an issue. It is also possible that the process of embryonic development that preceded parasitism increased accessibility of various nutrients within the host egg for the parasitoid. Given that the parasitoid's size was unaffected and that its development was accelerated in fertilized eggs, it would seem that fertilized eggs are nutritionally of higher quality.
Lines 299-300: Optimal egg age for parasitization also may be an issue. It would be very useful to know the age(s) of host eggs used in the experiment.
Line 307-308: This may or may not be true. I would still lean toward developmental effects in fertilized eggs that improve accessibility to nutrients in the host for the parasitoid. How does the suitability of unfertilized and fertilized eggs for parasitism change over time? I would expect the suitability of fertilized eggs to decline as the embryo develops, probably more rapidly than a decline in the unfertilized eggs.
Lines 332-333: I would imagine that there is also quite a bit of work to be done on how to extract the eggs efficiently for mass production. Unless A. pernyi is entirely pro-ovigenic, killing the moth to remove eggs from it will also reduce the lifetime fecundity of the moth for producing eggs for mass rearing. What is the cost involved in producing a batch of, say, 100 eggs from fertilized females allowed to oviposit throughout their lives versus unfertilized females killed to harvest eggs? How labor intensive are the two approaches?

Author Response
Reviewer 2:
The manuscript is interesting and generally clearly written. The results may become very useful in developing a mass-rearing program. I made a variety of edits on the manuscript to improve clarity, and I hope the authors find those useful. I did not review the literature cited in depth to make sure all were cited, or formatted correctly.
Response: Thanks for your comments on our study.
I also have some questions/comments about the manuscript (these are also in the comments on the pdf):
Line 28: It would useful to outline the treatments in the abstract, rather than making the reader try to extract them from the results text.
Response: Thanks for your suggestion. We add the treatments in detail in the abstract as you suggested after “five different treatments”.
Lines 67-69: I don’t understand this sentence. Can you clarify your points?
Response: We revised the text to ‘Although biocontrol programs are promising, low abundance of natural enemies were not sufficient to control the pest in biological control programs, and that there is a need for improving mass-rearing techniques for field augmentative release of natural enemies.’
Lines 122-123: How long after manual removal were eggs exposed to the parasitoids? Similarly, how long after washing were eggs exposed, and what was the post-oviposition age of the naturally deposited eggs at the time they were exposed to the parasitoids? What are the optimal host egg ages for parasitism by this parasitoid?
Response: According to the comment, we revise these sentences as ‘The MUW eggs were collected by dissecting the abdomen of unmated and mature female moths and extracted eggs were subject to washing with distilled water immediately. Then washed eggs were air dried under the room temperature (about 1 h). When eggs dried, immature green eggs were removed, and the healthy eggs within 4 h after drying were used for experiment’. 0-day-old naturally eggs were exposed to the parasitoids. The optimal host egg ages was not test for this parasitoid, the current study focused on optimizing extraction treatments of fertilized or unfertilized, the optimal host egg ages may be tested in future study when optimal extraction treatments are determined. So, this study we only test 0-day-old eggs.
Lines 122-131: How did egg size compare within the manually extracted eggs? Were they all similar in size? And how did the manually extracted egg size compare with eggs naturally oviposited?
Response: Egg diameter was used to compare egg size. Both manually extracted eggs and naturally eggs showed similar egg diameter.
Lines 167-169: Were there any efforts made to assess number of parasitoids per parasitized egg to see if the parasitoids modified their oviposition patterns in response to the choice options?
Response: Unfortunately, we did not assess number of parasitoids per parasitized egg in the choice test. Thanks for your suggestion, we may consider testing this in future studies.
Line 237: There were only significant differences in % female offspring between MUW and NUW.
Response: Thank you for pointing out, it was a typo. We revised to “There was significant differences in percentages of female progeny between MUW eggs and NUW eggs.”
Lines 280-282: How was this softening recognized? Did you measure behavioral elements of oviposition to identify this?
Response: Friction possibly soften egg chorion, which is just a speculation. Further studies will be needed to justify this. The word “observed” is changed to “assumed”. In the experiment, we found that parasitoid spend more time oviposit on naturally eggs. But behavioral elements of oviposition may be needed to confirm.
Lines 295-297: This idea depends on the status of the host egg after oviposition. If the parasitoid essentially stops development after oviposition, as is common among egg parasitoids, then competition is not an issue. It is also possible that the process of embryonic development that preceded parasitism increased accessibility of various nutrients within the host egg for the parasitoid. Given that the parasitoid's size was unaffected and that its development was accelerated in fertilized eggs, it would seem that fertilized eggs are nutritionally of higher quality.
Response: According to your suggestions, this sentence is revised as to “the process of embryonic development that preceded parasitism increased accessibility of various nutrients within the host egg for the parasitoid.”
Lines 299-300: Optimal egg age for parasitization also may be an issue. It would be very useful to know the age(s) of host eggs used in the experiment.
Response: Thanks. It is a good suggestion, we will evaluate the effect of host egg age on parasitization in future.
Line 307-308: This may or may not be true. I would still lean toward developmental effects in fertilized eggs that improve accessibility to nutrients in the host for the parasitoid. How does the suitability of unfertilized and fertilized eggs for parasitism change over time? I would expect the suitability of fertilized eggs to decline as the embryo develops, probably more rapidly than a decline in the unfertilized eggs.
Response: According to your suggestions, this sentence is revised as “embryonic development of fertilized A. pernyi egg improve accessibility to nutrients in the host egg for the parasitoid”
Lines 332-333: I would imagine that there is also quite a bit of work to be done on how to extract the eggs efficiently for mass production. Unless A. pernyi is entirely pro-ovigenic, killing the moth to remove eggs from it will also reduce the lifetime fecundity of the moth for producing eggs for mass rearing. What is the cost involved in producing a batch of, say, 100 eggs from fertilized females allowed to oviposit throughout their lives versus unfertilized females killed to harvest eggs? How labor intensive are the two approaches?
Response: A. pernyi is entirely a pro-ovigenic moth. The egg load will be same and not be increased as soon as female moth emerge. In our study we observed that the number of eggs obtained manually and naturally deposited is similar. Although we did not dive into cost of production, in China, the cost is low with labor and therefore the manually extracted eggs are easy to collect. Hope this will answer your question.
Round 2
Reviewer 1 Report
Review for Insects
11th June, 2021
Parasitism and suitability of Aprostocetus brevipedicellus on Chinese oak silkworm, Antheraea pernyi, a dominant factitious host.
This is a worthwhile study investigating the most optimal A. pernyi egg treatment for mass rearing of A. brevipedicellus. The manuscript has been considerably improved by reducing the number of detailed references to others work and improving some of the English used. The manuscript still needs major revision to further improve the English used and removing figures for insignificant results.
Line numbers below are from the “accept all changes version” of the manuscript and not the version with tracked changes.
Simple Summary:
The last sentence is too long and needs to be divided into two sentences.
Line 19: Need to explain what “better performance” means. Suggest to change to “… exhibited optimal parasitism on manually-extracted….”
Line 20: The abbreviation MUW is used here. Why wasn’t this used in the previous sentence after “manually-extracted A. pernyi eggs”? I think these are the same treatment?
Line 20: Need to explain what “showed better suitability” means. Suggest to state that “A. brevipedicellus offspring emerging from MUW eggs had high egg load suggesting that MUW eggs are optimal for mass production of A. brevipedicellus.
Line 21: The words “… to augment A. brevipedicellus for mass production” doesn’t make sense. Suggest to change to “A. brevipedicellus offspring emerging from MUW eggs had high egg load suggesting that MUW eggs are optimal for mass production of A. brevipedicellus.
Abstract:
Line 27: We cannot say “parasitism of A. brevipedicellus” – to use the phrase “parasitism of _____”, the blank needs to be the host. Change to “Here we evaluated A. brevipedicellus parasitism and fitness of their offspring on A. pernyi eggs with five different….”
Line 32: Suggest to include “significantly” before “preferred”.
Line 33: Change to “The pre-emergence time of parasitoid offspring emerging from fertilized eggs was shorter than….”
Line 34: Change to “…. more parasitoid offspring emerged….”
Line 36: Change to “The egg load of female parasitoid offspring emerging from MUW and NUW eggs was 30-60% higher than the remaining treatments”.
Line 37: Suggest to change to “Overall, MUW A. pernyi eggs are…..”
Introduction
Line 45: Need to include “in” before “11.4282 million ha”
Line 46: “and biodiversity” does not make sense here. Suggest to remove.
Line 49: There are several mistakes in this sentence. Please change to “Furthermore, pesticide applications negatively impact non-target organisms such as beneficial arthropods (e.g. natural enemies of pests), lead to resurgence of secondary pests, reduce biodiversity, and impact overall ecosystem sustainability.”
Line 55: Please remove “etc”.
Line 57: This sentence is difficult to understand. Suggest to change to “Although biocontrol programs are promising, low abundance of natural enemies can lead to failure of biological control programs, demonstrating that there is a need for….”
Line 65 and Line 67: Both sentences start with “currently”. Suggest to change the second one to “To our knowledge, no mass rearing of….”
Line 68: Insert commas after “silkworm” and “Meneville”.
Line 69: “rearing efficiency as host” and “lower cost and easy transport” does not make sense. Suggest to change to “….. such as Trichogramma and Anastatus because they are simple and cheap to mass produce, and easy to transport.”
Line 70: Why are they easy to transport? I’m not sure what this means. Please explain.
Line 71: This sentence is confusing. Suggest to change to “A recent study demonstrated that Trichogramma parasitoids prefer parasitizing manually-extracted unfertilized washed A pernyi eggs, and Trichogramma offspring had increased fitness when reared on this factious host compared to natural hosts.”
Line 75: Regarding “Previous studies have demonstrated that the fertilization status of host eggs”, this idea has already been talked about in the previous sentences because an example using Trichrogramma was given. Therefore, this is repeated idea. Lines 70-76 need to be restructured. Suggest to structure it in the following order:
- Sentence starting “Previous studies have demonstrated that the fertilization status…”
- “Generally parasitoids prefer ovipositing in fertilized host eggs, however, this is not always the case”
- Sentence starting “The extracted eggs of A. pernyi from unfertilized females…”
- Sentence starting “A recent study demonstrated….”
- Sentence starting “However, whether these unfertilized eggs can be used….”
Line 77: Suggest to change “Also” to Additionally”
Line 80: Change to “The objective of the current study….”
Methods:
Thank you for your response to the use of 10% versus 15% honey water. I suggest to include this in the material and methods.
Line 119: Suggest to change “so” to “therefore”.
Line 124: Suggest to change to “The host eggs were then cut out from the card and held individually….”
Line 126: Suggest to change to “All parasitized eggs were checked daily until emergence of all adults. After there was no further adult emergence, host eggs were…..”
Line 136: How many replicates of the no choice treatment were set up? Were there 30 replicates tested for each treatment?
Line 137: There needs to be a space between “load” and “per”
Line 141: Change “…to A. pernyi eggs…” to “…for A. pernyi eggs…”.
Line 141: Change “previous” to “previously”
Line 148: Regarding “For each egg treatment, 30 female A. brev were tested”. All treatments were in one vial for the Choice experiment. Maybe you meant to have this sentence for the No Choice experiment? For the Choice experiment, you could state that “This experiment was replicated 30 times.”
Line 150-153: Suggest to change to “For the no choice bioassay, a one-way analysis of variance (ANOVA) was conducted to determine the effect of treatment on the number of parasitized…….and egg load of female offspring. Tukey’s honestly significant difference …..”
Line 157: Suggest to change to “…. and washing treatments on the various parameters of A. brevipedicullus previously listed for the no-choice bioassay.”
Line 158: Please explain why Friedman non-parametric analyses was used for the choice test and not ANOVA like the no-choice test. Did data not meet normality assumptions of ANOVA? What did the Friedman non-parametric analyses test? The effect of ? on ?.
Results:
Lines 163 and 167: Please keep results for one variable together. Suggest to change to “….. A. pernyi eggs in all egg treatments, and there was a significant difference in the number of parasitized eggs among treatments (F = , P = ). The manually-extracted egg treatment resulted in maximum number of parasitized eggs which was 63%-242% higher than the remaining treatments. Fertilization significantly affected the…….”
The percentages on Line 168 can be rounded off to 63%-242%.
Line 169-175: Suggest to report the most significant/main results first. Therefore, report results from MUW first. Is there a reason that you write out “manually-extracted egg treatment”, but use abbreviations for NFW, NUW, NFUW and NUUW? Also, instead of state “more”, say how much more. Suggest to change to “In choice tests, MUW A. pernyi eggs resulted in maximum number of parasitized eggs, which was 259%-2533% higher than the remaining treatments. A brevipedicellus parasitized ?%more NUW eggs than NUUW, and parasitized ?% more NFW eggs than NFUW eggs. The number of parasitized eggs was equivalent between NUUW and NFUW treatments.
Line 189: Aprostocetus doesn’t need to be in full.
Line 192: Instead of “significantly shorter”, write how much % shorter.
Line 193: You have reported maximum and minimum which can be easily seen from the figure. Suggest to report a percentage difference, for example change to “Overall, development of A. brevipedicuellus offspring was ?% -?% shorter on NFW and NFUW treatments compared with MUW, NUW and NUUW treatments.”
Line 201: Aprostocetus doesn’t need to be in full.
Line 202: “or” needs to be changed to “and”
Line 203-204: You have reported the ranking of the treatments which can be easily seen from the figures. Suggest to report a percentage difference, for example, “The number of emerged adults per egg was ?%-?% higher in the NFUW treatment compared with the remaining egg treatments. MUW and NUW egg treatments resulted in about 7 adults per host, which was significantly lower than the remaining treatments.”
Lines 205-212: This section is confusing. Suggest to start with the effect of treatment on the various parameters like you have done for the previous section, then shift to the effect of fertilization and washing on the various parameters.
Line 207: Change to “Similarly, the number of dead parasitoids per host…..”
Line 209-210: There were no signicant differences in emergence rate and number of dead parasitoids per host…… between what? Is this sentence referring to the effect of treatment on these parameters?
Line 213: Change to “Treatment had a significant effect on the percentage of female progeny (F =, P = ). A. brevipedicellus offspring emerging from MUW eggs was ?% more female biased compared with the NUW treatment.”
Line 224 – 232: There needs to be consistency between sections on how you report the results. Suggest to start with the range (eg., average length of hind tibia) and then move into the significant differences for treatment, then the effect of fertilization and washing. Keep this consistent between sections.
Line 229: Change “or washing” to “and washing”.
Line 230-232: Listing the order from highest to lowest and including means isn’t very interesting or helpful because this can be easily seen from the figure. Instead, explain what significant differences there are and the percentage difference.
Figures:
Figure 1: It was difficult to see the “A” and “B” embedded in the figures to show whether they are Choice or No Choice. Suggest to move these below and to the left of the figures, or as journal requires.
Figure 2: Please list the five treatment as you have done for Figure 1 since so this figure can stand alone.
Figure 3: Please remove insignificant results – the figures for emergence rate and the number of dead parasitoids per egg need to be removed. Report the range in these parameters in the text. Please list the five treatment as you have done for Figure 1 since so this figure can stand alone.
Figure 4: HTL results are not significant and need to be removed from being expressed in a figure. Simply state in the results section the range in HTL and the test statistics showing they are insignificant. E.g., HTL ranged from 430-415 um ± SE and did not significantly vary between egg treatments (F = ? , P = ? ).
The figures can be combined into two figures. One figure for choice and no choice results. The other figure can contain the significant parameters for offspring: pre-emergence time, number of emerged adults per egg and percentage female progeny. If you are reducing the size of the individual figures to place into one figure containing three variables then the font needs to be increased so it’s easily read.
Discussion
Thank you for explaining in your response to reviewers why you didn’t include “manually-extracted unfertilized unwashed eggs”? Being able to use manually-extracted unfertilized unwashed eggs sounds like it would reduce the time needed to prepare eggs used for mass production of A. brevipedicellus. Your explanation could be included in the discussion when you talk about pitfalls or improvements of your experimental design and future research.
Line 248: “parasitism of parasitoids” isn’t correct. Suggest to use “parasitism rate”. Suggest to remove this sentence and incorporate this idea into the following sentence.
Line 249: Change to “Previous studies have demonstrated that host quality (site references) and fertilization of host eggs can affect parasitism rate, with most species preferring to parasitize fertilized eggs (site refernces).
Lines 251-252: There are too many commas separating connecting ideas that need to remain connected. Suggest to change to “Similar findings were also reported for T. japonicas and T. densdrolimi, which prefer unfertilized H. halys and A. peryni eggs for massing rearing.
Line 255: Change “The reason can be due” to “This may be attributable”
Line 256: Suggest to change to “Furthermore, the washing treatment may soften and thin the egg chorion due to friction between the eggs during washing. However, further studies are required to verify this phenomenon.”
Line 260: Are the references 33 and 34 in relation to studies that have shown that a hard egg chorion is not conducive to mass production? Or is this part of the sentence suggested by the authors because of the results from their own study. Maybe these references need to be placed like so: “Previous studies have demonstrated that the hardness and thickness of A. pernyi chorion is a limiting factor for some parasitoids (33, 34). Results presented here showed that a hard egg chorion can be an obstacle for parasitism and is not conducive to mass production.”?
Line 266: Suggest to change “on fertilized eggs” to “developing in fertilized eggs”.
Lines 269-275: The English isn’t quite right in this paragraph. Suggest to change to “The embryonic development of fertilized host eggs can significantly influence the fitness of parasitoids. The difference in the pre-emergence time between fertilized and unfertilized eggs seen in the current study may be attributable to the process of embryonic development, which increased accessibility of various nutrients within the host egg, before exposure to the parasitoid. This increase in nutrient availability may have led to faster development of A. brevipedicellus offspring on fertilized eggs than that on unfertilized eggs. Other factors may also be involved, e.g. egg deposition period (47), but they were not measured in the present study”.
Lines 276-280: This paragraph discusses the effect of fertilization and washing treatments on the number of emerged adults per host egg. The reason given for higher numbers of parasitoids emerging from fertilized eggs, was nutrition. The effect of fertilization needs to linked into the previous paragraph, and then the reason for (1) the shorter developmental time and (2) higher number of offspring per egg can be both explained by increase in nutrition. Otherwise it sounds like a repeated idea.
Line 283: Change “For host eggs with washing treatments” to “For the host egg washing treatments”
Line 285: Change possibly to “may”
Line 285-287: I’m not sure what this sentence means and how it relates to the previous sentence.
Line 293: Reference 60 can be placed with “58, 59, 60” and this example removed.
Line 294-295: Non-significant results should not be mentioned in the discussion. P = 0.53 so there were no significant differences in body size between treatments. You cannot say that MUW resulted in larger parasitoids.
Line 296-297: Suggest to change to “Female offspring emerging from MUW and NUW eggs had the highest egg load. NUW resulted in 30%-50% less parasitism compared with the MUW treatment suggesting that MUW A. pernyi eggs are the most optimal host for mass rearing of A. brev.”
Line 299: Suggest to change conclusion to “MUW eggs of A. pernyi were most sutiable for oviposition based on host preference, parasitism and parameters of offspring fitness.”
Line 303: Remove the “the” before forest pests.
Have previous studies shown a link between a lower number of offspring per host and a higher egg load? I would imagine so. Some discussion and literature would be helpful.
Author Response
Firstly, we are grateful for your valuable comments and revision suggestions on our paper. Under your instructions, the quality of our paper has been solidly improved. Please check our responses on your comments and revisions as followed.
Simple Summary:
The last sentence is too long and needs to be divided into two sentences.
Response: Thank you very much. We change the last sentence to ‘Among the host egg treatments, A. brevipedicellus exhibited optimal parasitism on manually-extracted, unfertilized and washed (MUW) eggs of A. pernyi, and A. brevipedicellus offspring emerging from MUW eggs had high egg load. The results indicate that MUW eggs are optimal for mass production of A. brevipedicellus.’
Line 19: Need to explain what “better performance” means. Suggest to change to “… exhibited optimal parasitism on manually-extracted….”
Response: Thank you very much. Done as suggested.
Line 20: The abbreviation MUW is used here. Why wasn’t this used in the previous sentence after “manually-extracted A. pernyi eggs”? I think these are the same treatment?
Response: Thank you very much. Done as suggested.
Line 20: Need to explain what “showed better suitability” means. Suggest to state that “A. brevipedicellus offspring emerging from MUW eggs had high egg load suggesting that MUW eggs are optimal for mass production of A. brevipedicellus.
Response: Thank you very much. Done as suggested.
Line 21: The words “… to augment A. brevipedicellus for mass production” doesn’t make sense. Suggest to change to “A. brevipedicellus offspring emerging from MUW eggs had high egg load suggesting that MUW eggs are optimal for mass production of A. brevipedicellus.
Response: Thank you very much. Done as suggested.
Abstract:
Line 27: We cannot say “parasitism of A. brevipedicellus” – to use the phrase “parasitism of _____”, the blank needs to be the host. Change to “Here we evaluated A. brevipedicellus parasitism and fitness of their offspring on A. pernyi eggs with five different….”
Response: Thank you very much. Done as suggested.
Line 32: Suggest to include “significantly” before “preferred”.
Response: Thank you very much. Done as suggested.
Line 33: Change to “The pre-emergence time of parasitoid offspring emerging from fertilized eggs was shorter than….”
Response: Thank you very much. Done as suggested.
Line 34: Change to “…. more parasitoid offspring emerged….”
Response: Thank you very much. Done as suggested.
Line 36: Change to “The egg load of female parasitoid offspring emerging from MUW and NUW eggs was 30-60% higher than the remaining treatments”.
Response: Thank you very much. Done as suggested.
Line 37: Suggest to change to “Overall, MUW A. pernyi eggs are…..”
Response: Thank you very much. Done as suggested.
Introduction
Line 45: Need to include “in” before “11.4282 million ha”
Response: Thank you very much. Done as suggested.
Line 46: “and biodiversity” does not make sense here. Suggest to remove.
Response: Thank you very much. Done as suggested.
Line 49: There are several mistakes in this sentence. Please change to “Furthermore, pesticide applications negatively impact non-target organisms such as beneficial arthropods (e.g. natural enemies of pests), lead to resurgence of secondary pests, reduce biodiversity, and impact overall ecosystem sustainability.”
Response: Thank you very much. Done as suggested.
Line 55: Please remove “etc”.
Response: Thank you very much. Done as suggested.
Line 57: This sentence is difficult to understand. Suggest to change to “Although biocontrol programs are promising, low abundance of natural enemies can lead to failure of biological control programs, demonstrating that there is a need for….”
Response: Thank you very much. Done as suggested.
Line 65 and Line 67: Both sentences start with “currently”. Suggest to change the second one to “To our knowledge, no mass rearing of….”
Response: Thank you very much. Done as suggested.
Line 68: Insert commas after “silkworm” and “Meneville”.
Response: Thank you very much. Done as suggested.
Line 69: “rearing efficiency as host” and “lower cost and easy transport” does not make sense. Suggest to change to “….. such as Trichogramma and Anastatus because they are simple and cheap to mass produce, and easy to transport.”
Response: Thank you very much. Done as suggested.
Line 70: Why are they easy to transport? I’m not sure what this means. Please explain.
Response: During the transportation, no worry on these eggs will be squeezed due to the hard chorin, sometimes also do not need specific storage conditions.
Line 71: This sentence is confusing. Suggest to change to “A recent study demonstrated that Trichogramma parasitoids prefer parasitizing manually-extracted unfertilized washed A pernyi eggs, and Trichogramma offspring had increased fitness when reared on this factious host compared to natural hosts.”
Response: Thank you very much. Done as suggested.
Line 75: Regarding “Previous studies have demonstrated that the fertilization status of host eggs”, this idea has already been talked about in the previous sentences because an example using Trichrogramma was given. Therefore, this is repeated idea.
Response: Thank you very much. According to your suggestions, these texts were restructured as followed.
Lines 70-76 need to be restructured. Suggest to structure it in the following order:
- Sentence starting “Previous studies have demonstrated that the fertilization status…”
- “Generally parasitoids prefer ovipositing in fertilized host eggs, however, this is not always the case”
- Sentence starting “The extracted eggs of A. pernyi from unfertilized females…”
- Sentence starting “A recent study demonstrated….”
- Sentence starting “However, whether these unfertilized eggs can be used….”
Response: Thank you very much. Done as suggested.
Line 77: Suggest to change “Also” to Additionally”
Response: Thank you very much. Done as suggested.
Line 80: Change to “The objective of the current study….”
Response: Thank you very much. Done as suggested.
Methods:
Thank you for your response to the use of 10% versus 15% honey water. I suggest to include this in the material and methods.
Response: Thank you very much. Done as suggested.
Line 119: Suggest to change “so” to “therefore”.
Response: Thank you very much. Done as suggested.
Line 124: Suggest to change to “The host eggs were then cut out from the card and held individually….”
Response: Thank you very much. Done as suggested.
Line 126: Suggest to change to “All parasitized eggs were checked daily until emergence of all adults. After there was no further adult emergence, host eggs were…..”
Response: Thank you very much. Done as suggested.
Line 136: How many replicates of the no choice treatment were set up? Were there 30 replicates tested for each treatment?
Response: There were 30 replicates for each treatment in the no choice experiment. We add the sentence “For each egg treatment, 30 female A. brevipedicellus were tested.” at the end of the paragraph.
Line 137: There needs to be a space between “load” and “per”
Response: Thank you very much. Done as suggested.
Line 141: Change “…to A. pernyi eggs…” to “…for A. pernyi eggs…”.
Response: Thank you very much. Done as suggested.
Line 141: Change “previous” to “previously”
Response: Thank you very much. Done as suggested.
Line 148: Regarding “For each egg treatment, 30 female A. brevipedicellus were tested”. All treatments were in one vial for the Choice experiment. Maybe you meant to have this sentence for the No Choice experiment? For the Choice experiment, you could state that “This experiment was replicated 30 times.”
Response: Thank you very much. Done as suggested.
Line 150-153: Suggest to change to “For the no choice bioassay, a one-way analysis of variance (ANOVA) was conducted to determine the effect of treatment on the number of parasitized…….and egg load of female offspring. Tukey’s honestly significant difference …..”
Response: Thank you very much. Done as suggested.
Line 157: Suggest to change to “…. and washing treatments on the various parameters of A. brevipedicullus previously listed for the no-choice bioassay.”
Response: Thank you very much. Based on your comments on result descriptions, we feel that it is not necessary to do a two-way ANOVA using a general linear model (GLM) to analyze the effects of fertilization and washing treatments on the various parameters of A. brevipedicellus in naturally-laid A. pernyi eggs. From the above one-way ANOVA, we also could found the main conclusion. In addition, as soon as we add the two-way ANOVA analyses in the results, in contrary, the readers could not easily find the main results. Finally, we decide to delete the two-way ANOVA and related result analyses.
Line 158: Please explain why Friedman non-parametric analyses was used for the choice test and not ANOVA like the no-choice test. Did data not meet normality assumptions of ANOVA? What did the Friedman non-parametric analyses test? The effect of ? on ?.
Response: For the choice test, the data did not meet normality assumptions of ANOVA. Therefore, we used Friedman non-parametric analyses referred to other similar experiments. The Friedman non-parametric analyses was conducted to analyze determine the effect of treatment on the number of parasitized eggs. We change the sentence “For choice test, data were analyzed using the Friedman non-parametric analysis.” to “For choice test, data were analyzed using the Friedman non-parametric analysis to determine the effect of treatment on the number of parasitized eggs.”.
Results:
Lines 163 and 167: Please keep results for one variable together. Suggest to change to “….. A. pernyi eggs in all egg treatments, and there was a significant difference in the number of parasitized eggs among treatments (F = , P = ). The manually-extracted egg treatment resulted in maximum number of parasitized eggs which was 63%-242% higher than the remaining treatments. Fertilization significantly affected the…….”
Response: Thank you very much. Done as suggested. We also delete the analysis results of two-way ANOVA.
The percentages on Line 168 can be rounded off to 63%-242%.
Response: Thank you very much. Done as suggested.
Line 169-175: Suggest to report the most significant/main results first. Therefore, report results from MUW first. Is there a reason that you write out “manually-extracted egg treatment”, but use abbreviations for NFW, NUW, NFUW and NUUW? Also, instead of state “more”, say how much more. Suggest to change to “In choice tests, MUW A. pernyi eggs resulted in maximum number of parasitized eggs, which was 259%-2533% higher than the remaining treatments. A brevipedicellus parasitized ?%more NUW eggs than NUUW, and parasitized ?% more NFW eggs than NFUW eggs. The number of parasitized eggs was equivalent between NUUW and NFUW treatments.
Response: Thank you very much. Done as suggested.
Line 189: Aprostocetus doesn’t need to be in full.
Response: Thank you very much. Done as suggested.
Line 192: Instead of “significantly shorter”, write how much % shorter.
Response: Thank you very much. Done as suggested.
Line 193: You have reported maximum and minimum which can be easily seen from the figure. Suggest to report a percentage difference, for example change to “Overall, development of A. brevipedicuellus offspring was ?% -?% shorter on NFW and NFUW treatments compared with MUW, NUW and NUUW treatments.”
Response: Thank you very much. Done as suggested.
Line 201: Aprostocetus doesn’t need to be in full.
Response: Thank you very much. Done as suggested.
Line 202: “or” needs to be changed to “and”
Response: Done.
Line 203-204: You have reported the ranking of the treatments which can be easily seen from the figures. Suggest to report a percentage difference, for example, “The number of emerged adults per egg was ?%-?% higher in the NFUW treatment compared with the remaining egg treatments. MUW and NUW egg treatments resulted in about 7 adults per host, which was significantly lower than the remaining treatments.”
Response: Thank you very much. Done as suggested. We also delete the analysis results of two-way ANOVA.
Lines 205-212: This section is confusing. Suggest to start with the effect of treatment on the various parameters like you have done for the previous section, then shift to the effect of fertilization and washing on the various parameters.
Response: Thank you very much. Done as suggested. We also delete the analysis results of two-way ANOVA.
Line 207: Change to “Similarly, the number of dead parasitoids per host…..”
Response: Thank you very much. Done as suggested.
Line 209-210: There were no signicant differences in emergence rate and number of dead parasitoids per host…… between what? Is this sentence referring to the effect of treatment on these parameters?
Response: Thank you very much. Done as suggested.
Line 213: Change to “Treatment had a significant effect on the percentage of female progeny (F =, P = ). A. brevipedicellus offspring emerging from MUW eggs was ?% more female biased compared with the NUW treatment.”
Response: Thank you very much. Done as suggested. We also delete the analysis results of two-way ANOVA.
Line 224 – 232: There needs to be consistency between sections on how you report the results. Suggest to start with the range (eg., average length of hind tibia) and then move into the significant differences for treatment, then the effect of fertilization and washing. Keep this consistent between sections.
Response: Thank you very much. Done as suggested.
Line 229: Change “or washing” to “and washing”.
Response: Done.
Line 230-232: Listing the order from highest to lowest and including means isn’t very interesting or helpful because this can be easily seen from the figure. Instead, explain what significant differences there are and the percentage difference.
Response: Thank you very much. Done as suggested.
Figures:
Figure 1: It was difficult to see the “A” and “B” embedded in the figures to show whether they are Choice or No Choice. Suggest to move these below and to the left of the figures, or as journal requires.
Response: Thank you very much. Done as suggested. We move the “A” and “B” top and to the left of the figures. Please check the updated figure.
Figure 2: Please list the five treatment as you have done for Figure 1 since so this figure can stand alone.
Response: Thank you very much. Done as suggested.
Figure 3: Please remove insignificant results – the figures for emergence rate and the number of dead parasitoids per egg need to be removed. Report the range in these parameters in the text. Please list the five treatment as you have done for Figure 1 since so this figure can stand alone.
Response: Thank you very much. Done as suggested. The sub-figures for emergence rate and the number of dead parasitoids per egg were removed, and the range of these parameters were reported in the text. We also list the five treatments as Figure 1.
Figure 4: HTL results are not significant and need to be removed from being expressed in a figure. Simply state in the results section the range in HTL and the test statistics showing they are insignificant. E.g., HTL ranged from 430-415 um ± SE and did not significantly vary between egg treatments (F = ? , P = ? ).
Response: Thank you very much. Done as suggested. We remove the subfigure of HTL results from figure 4. The description of the results changed as suggested.
The figures can be combined into two figures. One figure for choice and no choice results. The other figure can contain the significant parameters for offspring: pre-emergence time, number of emerged adults per egg and percentage female progeny. If you are reducing the size of the individual figures to place into one figure containing three variables then the font needs to be increased so it’s easily read.
Response: Thank you very much. Done as suggested. Finally, we retain three figures, combine the figure 2 and figure 3 into one figure. Please check the updated figures.
Discussion
Thank you for explaining in your response to reviewers why you didn’t include “manually-extracted unfertilized unwashed eggs”? Being able to use manually-extracted unfertilized unwashed eggs sounds like it would reduce the time needed to prepare eggs used for mass production of A. brevipedicellus. Your explanation could be included in the discussion when you talk about pitfalls or improvements of your experimental design and future research.
Response: Thank you very much. The manually-extracted, unfertilized and unwashed eggs require longer drying time, meanwhile, the eggs were covered by a layer of muscus, and all eggs were stick together, which makes it difficult to separate, it will spend more time and labor. Overall, manually-extracted, unfertilized and unwashed eggs would not reduce save the time if they will be used. Therefore, anually-extracted, unfertilized and unwashed eggs were not considered in the experiment.
Line 248: “parasitism of parasitoids” isn’t correct. Suggest to use “parasitism rate”. Suggest to remove this sentence and incorporate this idea into the following sentence.
Response: Thank you very much. Done as suggested.
Line 249: Change to “Previous studies have demonstrated that host quality (site references) and fertilization of host eggs can affect parasitism rate, with most species preferring to parasitize fertilized eggs (site refernces).
Response: Thank you very much. Done as suggested.
Lines 251-252: There are too many commas separating connecting ideas that need to remain connected. Suggest to change to “Similar findings were also reported for T. japonicas and T. densdrolimi, which prefer unfertilized H. halys and A. peryni eggs for massing rearing.
Response: Thank you very much. Done as suggested.
Line 255: Change “The reason can be due” to “This may be attributable”
Response: Thank you very much. Done as suggested.
Line 256: Suggest to change to “Furthermore, the washing treatment may soften and thin the egg chorion due to friction between the eggs during washing. However, further studies are required to verify this phenomenon.”
Response: Thank you very much. Done as suggested.
Line 260: Are the references 33 and 34 in relation to studies that have shown that a hard egg chorion is not conducive to mass production? Or is this part of the sentence suggested by the authors because of the results from their own study. Maybe these references need to be placed like so: “Previous studies have demonstrated that the hardness and thickness of A. pernyi chorion is a limiting factor for some parasitoids (33, 34). Results presented here showed that a hard egg chorion can be an obstacle for parasitism and is not conducive to mass production.”?
Response: Thank you very much. Done as suggested. The references 33 and 34 are in relation to studies that have shown that a hard egg chorion is not conducive to mass production.
Line 266: Suggest to change “on fertilized eggs” to “developing in fertilized eggs”.
Response: Thank you very much. Done as suggested.
Lines 269-275: The English isn’t quite right in this paragraph. Suggest to change to “The embryonic development of fertilized host eggs can significantly influence the fitness of parasitoids. The difference in the pre-emergence time between fertilized and unfertilized eggs seen in the current study may be attributable to the process of embryonic development, which increased accessibility of various nutrients within the host egg, before exposure to the parasitoid. This increase in nutrient availability may have led to faster development of A. brevipedicellus offspring on fertilized eggs than that on unfertilized eggs. Other factors may also be involved, e.g. egg deposition period (47), but they were not measured in the present study”.
Response: Thank you very much. Done as suggested.
Lines 276-280: This paragraph discusses the effect of fertilization and washing treatments on the number of emerged adults per host egg. The reason given for higher numbers of parasitoids emerging from fertilized eggs, was nutrition. The effect of fertilization needs to linked into the previous paragraph, and then the reason for (1) the shorter developmental time and (2) higher number of offspring per egg can be both explained by increase in nutrition. Otherwise it sounds like a repeated idea.
Response: Thank you very much. Based on your comments, we revise these sentences to ‘This is likely due to the increased accessibility of various nutrients associated with the fertilized A. pernyi eggs. We suspect that embryonic development of fertilized A. pernyi egg improve accessibility to nutrients in the host egg for the parasitoid. Therefore, when parasitizing fertilized eggs, the parasitoids might lay the larger number of eggs on per host egg. A recent study on Trichogramma parasitizing A. pernyi eggs also indicated that the number of emerged adults per fertilized host egg was signficantly higher than that per unfertilized host egg [34]. For the host egg washing treatments, there was significantly higher number of emerged adults per egg on unwashed eggs than that on washed. We suspect that, compared with the wahsed eggs, it is difficult for parasitoids to parasitize unwashed eggs with thicker and harder chorion. Therefore, if successfully parasitize a unwashed egg, parasitoids will expect to lay the larger number of offspring inside. This may be a parasitic strategy, when parasitoids are confronted with disadvantage conditions [50-51].
Line 283: Change “For host eggs with washing treatments” to “For the host egg washing treatments”
Response: Thank you very much. Done as suggested.
Line 285: Change possibly to “may”
Response: Thank you very much. Done as suggested.
Line 285-287: I’m not sure what this sentence means and how it relates to the previous sentence.
Response: We suspect that when parasitizing unwashed eggs which may have thicker and harder chorion, parasitoids may hard to oviposit. Therefore, if successfully parasitize an egg, parasitoids will lay the larger number of offspring. This may be a parasitic strategy, when parasitoids confront with disadvantage conditions.
Line 293: Reference 60 can be placed with “58, 59, 60” and this example removed.
Response: Thank you very much. Done as suggested.
Line 294-295: Non-significant results should not be mentioned in the discussion. P = 0.53 so there were no significant differences in body size between treatments. You cannot say that MUW resulted in larger parasitoids.
Response: Thank you very much. Based on your comments, we change the writings as followed.
Line 296-297: Suggest to change to “Female offspring emerging from MUW and NUW eggs had the highest egg load. NUW resulted in 30%-50% less parasitism compared with the MUW treatment suggesting that MUW A. pernyi eggs are the most optimal host for mass rearing of A. brev.”
Response: Thank you very much. Based on your comments, we change these sentences to ‘Although there were no significant differences in parasitoid body size between treatments, female offspring emerging from MUW and NUW eggs had the highest egg load. NUW resulted in 39%-61% less parasitism compared with the MUW treatment suggesting that MUW A. pernyi eggs are the most optimal host for mass rearing of A. brevipedicellus.’
Line 299: Suggest to change conclusion to “MUW eggs of A. pernyi were most sutiable for oviposition based on host preference, parasitism and parameters of offspring fitness.”
Response: Thank you very much. Done as suggested.
Line 303: Remove the “the” before forest pests.
Response: Thank you very much. Done as suggested.
Have previous studies shown a link between a lower number of offspring per host and a higher egg load? I would imagine so. Some discussion and literature would be helpful.
Response: Thank you very much. According to your suggestion, we add a case to support your comments ‘Li et al. reported that the number of emerged Oomyzus sokolowskii Kurdjumov adults per host was negative correlation with egg load.’